# Investigating Understandings of Antibiotics and Antimicrobial Resistance in Diverse Ethnic Communities in Australia: Findings from a Qualitative Study

**DOI:** 10.3390/antibiotics8030135

**Published:** 2019-09-02

**Authors:** Andrea Whittaker, Davina Lohm, Chris Lemoh, Allen C. Cheng, Mark Davis

**Affiliations:** 1School of Social Sciences, Monash University, Melbourne 3800, Australia; 2School of Clinical Sciences, Monash University, Melbourne 3800, Australia; 3Monash Infectious Diseases, Melbourne 3168, Australia; 4School of Public Health and Preventive Medicine, Monash University, Melbourne 3800, Australia; 5Infection Prevention and Healthcare Epidemiology Unit, Alfred Health, Melbourne 3181, Australia

**Keywords:** antimicrobial resistance, ethnicity, antibiotic, qualitative, lay understandings

## Abstract

This paper explores the understandings of antibiotics and antimicrobial resistance (AMR) among ethnically diverse informants in Melbourne, Australia. A total of 31 face-to-face semi-structured qualitative interviews were conducted with a sample of ethnic in-patients who were admitted with an acquired antimicrobial infection in a public hospital (*n* = 7); five hospital interpreters; and ethnic members of the general community (*n* = 19) as part of a broader study of lay understandings of AMR. Thematic analysis revealed there was poor understanding of AMR, even among informants being treated for AMR infections. Causes of the increasing incidence of AMR were attributed to: weather fluctuations and climate change; a lack of environmental cleanliness; and the arrival of new migrant groups. Asian informants emphasized the need for humoral balance. Antibiotics were viewed as ‘strong’ medicines that could potentially disrupt this balance and weaken the body. Travel back to countries of origin sometimes involved the use of medical services and informants noted that some community members imported antibiotics from overseas. Most used the internet and social media to source health information. There is a lack of information in their own languages. More attention needs to be given to migrant communities who are vulnerable to the development, transmission and infection with resistant bacteria to inform future interventions.

## 1. Introduction

Antimicrobial resistance (AMR) is recognized as a global health emergency. AMR is a multifaceted health problem which compromises surgery, the management of chronic diseases, and the treatment and prevention of infectious diseases. Multi-drug resistant *Staphylococcus aureus* (MRSA), carbapenemase-producing enterobacteriaceae (CPE), multi-drug resistant tuberculosis (MDR-TB), are now all difficult to treat, creating problems for individual patients, their treating physicians, and health systems [1]. In an effort to meet the AMR challenge, global public health systems [2] are improving the surveillance of AMR in human and animal populations and rationalizing the prescription and dosing of antimicrobials in hospitals [3], general practice [4,5,6], and aged care [7]. The prevention of AMR is commonly cast as ‘stewardship’ [8] or ‘rational’ prescribing [9].

In Australia antibiotic-resistant gram-negative bacterial infections were once thought to be “hospital-acquired infections”, but people with community-acquired multiresistant gram-negative bacterial infections are now presenting to general practices and emergency departments [10]. Some of the most common community-acquired infections are becoming more difficult to treat. The most recent report on antimicrobial resistance surveillance in Australia indicates that *Escherichia coli* resistances to common agents has continued to rise in community-onset infections as well as ciprofloxacin resistance in *Salmonella* (which exceeded 60% in 2017) and community-associated methicillin-resistant *Staphylococcus aureus* has become highly prevalent in remote regions [10,11]. As yet the prevalence of these isolates within various migrant groups in Australia is unknown.

Australia has a diverse population, with an increasing number of permanent and temporary visas given to voluntary migrants and humanitarian settlers each year. Over seven million migrants from a wide range of countries and cultural backgrounds were living in Australia in 2018, just over 29 per cent of Australia’s resident population [12]. It is important to elicit the understandings of AMR from diverse backgrounds as studies in other settings note the importance of ethnicity to knowledge, practice and awareness of appropriate antibiotic use and AMR [13,14,15,16]. In addition, migrant populations are especially vulnerable: their transnational mobility means that members of these communities regularly return to their countries of origin for extended periods making AMR transfer more likely, or may already be colonized with resistant bacteria in their gut with the possibility of spread; newly arrived migrant groups may come from countries with health systems that may use antibiotics with less regulation; and current AMR communications which are largely disseminated in English are unlikely to reach these communities due to linguistic barriers.

Current models of how AMR originates, is sustained in the community or transfers between members of the community remains a source of debate within the scientific community. There are many potential sources of multiresistant organisms. These include transmission from other individuals (whether in the community, in institutions or in healthcare facilities), animals or food, the environment or from antibiotic consumption. However, the relative contributions of these is not known, and probably varies between different bacteria [17,18]. However, international travel is a well-established mode of global dissemination of multiresistant microorganisms [2,19]. For example, Carbapenem-resistant enterobacteriaceae (CRE) originated in the United States, spread to Mediterranean countries and then to the rest of the world, and bacteria carrying the enzyme New Delhi metallo-*β*-lactamase (NDM-1) which originated in India, have since been introduced to other countries including the United States and the United Kingdom [2]. Travel to areas with high rates of MDR Gram negative bacteria or healthcare exposure in these countries are listed as a major high-risk factors for AMR infection [20]. For example, rates of resistance to third-generation cephalosporins and fluoroquinolones in the United States, Southern Europe and much of Asia is rising [10].

In Australia there is a growing burden of multidrug-resistant infections among returned Australian travelers [21]. Epidemiological studies note the increasing rates and clinical consequences of nalidixic acid-resistant isolates causing enteric fever in returned travelers [22] and the role of international travel in colonization with *Escherichia coli* resistant to “critically important” antibiotics [23]. Ten cases with MDR gram-negative organisms were described among patients either repatriated directly to Australia or discharged from hospitals overseas following accidents causing orthopedic fractures [21]. To date, a small number of cases carrying New Delhi metallo-*β*-lactamase (NDM-1) have been reported in adults in Australia [10]. In all cases, patients travelled to the Indian subcontinent; many required hospitalization for their infection.

Research around the world suggests that lay people carry a range of mistaken understandings about AMR and this is also true of the Australian population. Survey data from many countries suggest that knowledge of AMR and correct antibiotic use varies by race, ethnic origin, sex, and educational status—factors that point to the importance of ensuring that educational campaigns reach underserved populations [24]. In Sweden, for example, a random household survey found that 26.8% of participants believed that antibiotics were effective against viruses and that 84.7% endorsed the mistaken idea that humans become resistant to antibiotics [25]. A comparative study of antibiotic knowledge in 11 European countries found variation in knowledge, most particularly with regard to inaccurate knowledge of antibiotic resistance, which ranged between 29% (Belgium) to 83% (Lithuania) of respondents [26]. In a study in Jordan [27], despite a high level of usage (41%) in the last two months, the majority (39–54%) of 1009 survey respondents believed in common misconceptions of antibiotics use and the majority (70%) did not know the term “antimicrobial resistance.” Little empirical data exist about the effect of national campaigns on public knowledge regarding antibiotics [24].

A systematic review of research from Europe, America and Asia (*n* = 54 articles) found that public knowledge of AMR was partial and mistaken [28], including beliefs that it: referred to personal immunity rather than microbial resistance; was a low risk for the individual; was caused by the action of others; and was a problem for health systems and not for the general public. A UK survey [29] found members of the general public were unclear about bacterial resistance and their explanations were generally incongruent with prevailing biomedical concepts. Few recognized resistant infections as a problem in the community. A qualitative interview study in nine European countries found people commonly attributed antibiotic resistance as a property of bodies rather than bacteria [30]. A review of behavioral perspectives on AMR for the UK’s Department of Health [31] found that knowledge was in general insufficient to support AMR interventions and that publics were confused by scientific assumptions. A focus-group study involving twenty-three Swedes [32] found that metaphors such as climate change were employed to describe antibiotic resistance as a slowly emerging problem. They reported a discrepancy between the risk of being personally affected by AMR and the perceived seriousness of a future without effective antibiotics and suggest that this poses a difficulty for the framing of AMR prevention.

As yet there is little research of the understandings concerning AMR or antimicrobial use within the general lay population in Australia [33] nor within ethnically diverse communities in Australia. Lum et al. [33] undertook structured interviews with thirty-two members of the public recruited from a university campus and found that their sample conceptualized AMR in four ways, (1) that one’s body becomes resistant to antibiotics; (2) that the medication is no longer effective; (3) that bacteria is becoming ‘stronger and resistant to the antibiotic’; and 4) that society as a collective whole is immune to antibiotics. Many of their informants struggled to adhere to their prescribed dosages, often failing to complete courses of treatment.

One Australian study involving migrants [34] recruited four hundred and seventeen Chinese participants in an online health survey involving a bilingual questionnaire. They found that around 24.2% of Chinese migrants used antibiotics without medical consultation for their last upper respiratory tract infection (URTI). Almost 70% of Chinese migrants indicated that they would stop taking antibiotics when symptoms improved; around 61% would use leftover antibiotics if they had similar symptoms. Almost half of Australian Chinese migrants had stored antibiotics at home [34]. A questionnaire survey of three hundred Indian, Egyptian, and Korean migrants in neighboring New Zealand [35] found low levels of understanding: most of those surveyed (73.3%) knew that antibiotics killed bacteria, but commonly also gave other incorrect responses. Almost half the sample (43.3%) believed colds and flu were caused by bacteria. Only 45.4% were sure that antibiotics were not useful for colds and flu.

The purpose of this paper is to describe the understandings and experiences of people from diverse ethnic backgrounds in Australia to antibiotic use and AMR captured through face to face qualitative interviews with community members to inform public antimicrobial stewardship programs and education. Throughout this paper we will refer to ‘diverse ethnic communities’ to refer to first generation migrant communities in Australia, also often described as ‘culturally and linguistically diverse’ (CALD) communities. We acknowledge these terminologies are problematic as they assume a mythic dominant homogenous Anglo-Australian culture against which all other ethnicities are defined. Also such a classification fails to acknowledge the various intersectional issues that cross-cut health vulnerabilities of migrant groups in Australia, such as length of time since arrival, refugee or migrant visa status, language use, knowledge of English, class, and occupation and education level. We need to stress that, as little is known about understandings and practices regarding AMR within the broader Australian community, we do not wish to single out or stigmatize migrants as particularly problematic or the cause of the spread of AMR in Australia. Rather, we wish to focus attention upon groups of people generally neglected by current approaches to the reduction of AMR in Australia and who may be particularly vulnerable to AMR infections.

## 2. Results

Major findings from the interviews are summarized in Table 1. They comprise a complex range of concepts and assumptions about the nature of antibiotics as powerful; humoral understandings of the body and illness; effects of mobility; the transformation of the climate; lack of hygiene; experience of different prescribing cultures; and notions of the resistant body. These explanations were sometimes used in combinations and suggest fluid and contingent understandings that are not necessarily tied to scientific concepts.

### 2.1. Impact of AMR Morbidity

Our interviews with migrant patients currently admitted in hospital reveal the impact of AMR complications upon the community. One infectious diseases doctor stated that in the hospital in which the study was conducted approximately every day they had around five or six patients with resistant infections. These patients suffered complications, long hospital stays and required long and complex treatments.

For example, Leona is a twenty-six year old diabetic Pacific Islander patient who presented with a small abscess in her side. Initial treatment failed. When we interviewed her she had spent six weeks in hospital:

“So they put me on a drip but after a week of having high fevers they decided that I was going to through theatre to have it cut open to try and bring all the ooze out. …they were just explaining that they tried to do their best but then apparently the Vac dressing they were putting me on wasn’t working. [She had another two surgeries] … it was quite massive and so deep they said it was 10 centimeters deep. So they cut it open and then I had to go to theatre a few more times to clean it out. So they had to take out a little muscle… (26 year old Pacific Islander, diabetic presented with abscess on side, 6 weeks in hospital).”

Similarly, Malia was hospitalized following a stroke. She then contracted an infection in hospital in the site of her intravenous line: “Yeah. I think this, the left arm, this arm it got, they said, “Oh, blood infection. It’s, it’s big. It’s swollen.” Different from my right side. Yes. And they do something. They put a tube in here. [Gestures] Yeah. Tube in here into my, my heart.” Due to her condition at the time she has little recollection of her time in hospital and could not recall what antibiotics she received. Several patients interviewed in the hospital were unaware of their resistant infection, they attributed their lengthy stays to complications but few could give accurate details of their illness or treatment.

Within the hospital patients may feel stigma due to the need for gloving and infection control protocols. One of the Dari interpreters noted that some patients do not understand why extra precautions may be taken:

“I have had quite a few [Afghan] patients that they have had infections and that we have to be careful and gown up so we don’t transfer it to other patients. And yeah they do look at you and go: “Why? You’re wearing all of that?’ And it’s kind of they feel a bit angry. ‘Are you isolating me? What’s going on? So when they explain to them, some they can understand and some can’t … I would say for CALD patients from Afghanistan, depending on their education, if the patient hasn’t got much education it’s very hard for them to understand [AMR].”

Doctors in the hospital setting were highly aware and concerned about AMR and faced daily decisions trying to deal with resistant infections. During grand rounds we heard discussions by orthopedic surgeons having to make decisions about amputation for a 27-year-old patient with an infected bone implant and the need to try to preserve as much bone as possible; infectious disease specialists described the difficulties at deciding about therapies especially when there were co-morbidities. For doctors and specialists in hospitals AMR figured as a constant threat and challenge.

### 2.2. Understandings of Antibiotics and AMR 

All migrants bring with them prior understandings and expectations of medicine use and prescribing knowledge. For example, Helena is a 36-year-old woman with a PhD and two young children who has lived in Australia for ten years. She avoids using antibiotics. She attributes this to her upbringing in Sri Lanka:

“I think I grew up without having taking a lot of antibiotics because my parents were into herbal medicine a lot. Like Ayurvedic medicine. Yeah. Because I clearly remember when I was a kid some of my family, like my, my mother’s sisters were into western medicine but my mum and my dad they were into Ayurvedic medicine. And then we hardly got sick compared to the other kids. And my mum and dad were telling them, “See, that’s what happens if you start to get … western medicine and antibiotics, and things like that … So next time you go to higher dosage and then you will be sick often because your resistance will be low.”

Instead she tends to use remedies she learned in Sri Lanka, such as the use of a tea from coriander and ginger to fight colds, or the insertion of peeled garlic in an infected ear:

“For example, for ear infections when we were little, we weren’t given antibiotics. The first thing they do is like they peel a garlic and then put it on your ear. … I remember like especially after swimming and things like that, getting a cold, and my parents putting that garlic on my, in my ear, and then feeling better the next day. And then you don’t need medicine at all. I do with my kids that.”

She uses these remedies without reference to any chemical properties of the herbs used, rather she simply refers to her past lived experience of efficacy.

Like Helena, many informants were reluctant to use antibiotics. Anik from India has lived in Australia 8 months and has a Masters degree. She stated:

“So, as far as possible, we try to keep ourselves away from antibiotics, try to manage it without … if we feel the normal medicine is not helping, then we go onto the antibiotics because they say that the body loses its ability to fight the infection; it’s not able to do it on its own. And you, like you’re abusing your body with too many antibiotics. So, as far as possible, try to keep it off. So we try the normal medicines’ especially with their children.”

Mothers in particular were inclined to try home remedies when their children were sick. Ginger and honey or lemon juice and honey, turmeric and pepper or turmeric in milk were offered before seeking any medical advice. Malika, a 36-year-old woman from Afghanistan who has been resident in Australia for three years following several years residence in New Zealand, avoided using antibiotics when possible, preferring to let her children fight infections without them and checking with a doctor. Mai, from Vietnam differentiated between ‘traditional’ Vietnamese medicines such as herbal treatments, inhalations and use of steam saunas to treat infections versus ‘western’ medicines. She always preferred to try ‘slower’ Vietnamese medicines as a first resort.

Other informants spoke of changing their practices regarding the use of antibiotics once they came to Australia. For example, Kia, thirty-five years old from Ethiopia who had lived in Australia for fifteen years, said that for treating her three children she used to be reluctant to use antibiotics preferring home remedies and when she did use them would not finish the whole course as she was uneasy about their use: ‘I didn’t know then. So I’ll leave it halfway and then the kid might feel better for two weeks, three. Then it come back again and then I think for not finishing the antibiotics has impact as well’.

She now knows the need to finish the entire prescribed course, but noted: ‘I’m not really fond of antibiotics’. Such descriptions suggest that antibiotic practices form part of household cultures and care-giving exchanges, pre-existing rationalities and practices. They also show that people carry a range of concerns about antibiotics regarding their strength or effects on their bodies and in many cases try to avoid using them. favoring household treatments as a first resort.

### 2.3. Humoral Understandings and Antibiotics

Humoral understandings of health remain common in many cultures, particularly across Asia and emerge in informants’ descriptions of sickness causality which is commonly attributed to sudden changes in body temperatures. For example, Azra, a 39-year-old Pakistani informant, described her children’s health as influenced by sudden changes in weather temperature or inappropriate consumption of ‘hot’ or ‘cold’ drinks. Here she is referring to an explanatory model of the body as influenced by humoral balance: ‘[They get sick] Just when weather will, going to change [Yeah] and they drink or eat something wrong like very cool water or cool drinks. So then they get some problems.’ Similarly, Hanifa from Afghanistan suggested that wearing too many clothes in winter was not helpful due to the sudden changes in temperatures the body receives when you change out of them. For this reason, she advocated not overdressing her children in winter. These humoral understandings also apply to understandings of antibiotics which are classified as ‘hot’ and strong’ medicines in many cultures. Such understandings exist alongside biomedical explanations and may require the ‘balancing’ of bodily humors when consuming antibiotics through the consumption of foods with ‘cooling’ properties. For example, Anik who is a trained nutritionist from India gave detailed scientifically correct descriptions of antibiotics and their uses, yet as she described various side effects of antibiotics noted, ‘you know, you need to take something very cooling with the antibiotic’. She stated: ‘when I was taking antibiotics, I made sure I had a lot of water, a lot of yoghurt and a lot of water because it’s supposed to be very cooling for the system’.

As a result, antibiotics are often viewed as powerful and potentially damaging. In many informants’ explanations, antibiotics were described as weakening the body or immune system, or damaging the body. For example, Azra, from Pakistan who had been resident for eleven years in Australia stated: “It’s my opinion because, when virus attack you, they just damage your organ outside. But, when you use some antibiotic, they damage your, inside your organs.” She viewed antibiotics as precipitating the strengthening of ‘viruses’: “Yes, I’m just, when, when we use harsh antibiotics, the virus is going to, when they come again, they are fighting very strongly again.” Similarly, Sadequa from Afghanistan explained, ’some antibiotics very strong, you know. If the person is weak, is, she doesn’t have enough energy to take these things, so how can she cope?’

Antibiotics were viewed as becoming ineffective for individual bodies, rather than resistant infections. Azra stated that due to the prolific prescribing of antibiotics in her home country of Pakistan, many no longer ‘worked on her body’ for her: ‘I’m just now [have] nine antibiotic[s that do] not work on my body. Yes. Just three antibiotics [work]. Now it’s working’. She described antibiotics as needed when the ‘viruses’ are strong but that the body needs time to attempt to cure an infection itself:

“Some doctor told us, “This is not good for your health because you give little time to, your fight with this virus a little bit means one week or maybe two weeks. But, if you think the virus is very strong, so you need antibiotic. [Right] Otherwise you, means you ignore antibiotic.”

A number of informants were either vague or uncertain about exactly what antibiotics were and what they are used for. For example, Preetha, who is a tertiary-educated 40-year-old Indian resident who has lived in Australia for the past eight years, was asked to describe an antibiotic. She replied it was ‘medicine’ used ‘sometimes [for] fever, sometimes cough, sometimes pain’. As in Azra’s case above, for many of our informants, bacteria and viruses were undifferentiated or used interchangeably. Likewise, understandings of the immune system varied, with some informants not knowing the term, but describing the body’s ability to ‘fight’ fevers or colds, the need to be ‘strong’ and ‘high’ [in level]. The use of antibiotics was both an indication of a lack of strength and in itself was understood by many informants as weakening the body. For example, Patricia, an elderly South African patient in hospital who had been hospitalized for over five weeks being treated for a resistant infection causing diverticulitis (inflammation of the lining of the intestine) described her body as usually containing infections, but that her immune system had been weakened by the use of antibiotics allowing the infections to develop:

“I didn’t know I had this infection and I didn’t think of infections before that. They say people usually have some infection inside them but it’s contained … it won’t spread or anything because it is contained in your body. But because of this diverticulitis and they found it, so I don’t know how it spread…because I’ve had it over a long period of time… My body has a strong immune system. … so the more you have antibiotics it sort of breaks down that immune system.”

### 2.4. Causes of the Rise of AMR 

When asked what they felt was causing the increase in ‘superbugs’ and AMR in Australia most informants stated they had ‘no idea’. Across the sample three important themes arose. The first was that of the Australian environment and weather as both being conducive to growing ‘bugs’ but also disturbing body humors due to constant temperature changes and extremes. The second theme was of the need for hygiene, both personal but also environmental. Third, ‘new’ migrants were seen as problematic, not only for carrying ‘bugs’ on their skin, but due to their different hygiene habits. Some like 50-year-old Aneesa, from Afghanistan, gave a multicausal explanation which combined changed weather conditions upsetting humoral body balance as well as blaming spread upon ‘new arrivals’ and unclean environments:

“To be honest, I think new arrivals coming about of a different country. You don’t see these things [bugs] until they settle or you know if might not be a cleaner place in Australia, you know. It’s a clean environment but also the weather changes, does affect our surroundings.”

“… They when they come on board from other countries, they need to do a thorough testing on them, humans as well ‘cause they bring them. … So they need to be checked thoroughly especially on the skin… and also if we see some places like lakes and all that, that needs, that needs maintenance. Like they need to check, clean those environments, like spray something around so the bugs don’t come about.”

In these quotes the common terminology of ‘bugs’ used to describe microbes seems to be confusing and associated with insects. Kia from Ethiopia likewise located the source of AMR in the Australian environment and migration. She stated:

“I think Australia is known for a lot of animals, there’s so many things… lizards and bugs and things. … I mean the geographical location of Australia, you know is an advantage for them [bugs] to breed, one thing. The other thing maybe also migration, you know. People come from lowland highland you know from Asian from everywhere… I mean Melbourne where I live now is very multicultural… it’s difficult even passing message across; it’s difficult because there’s second language speaking people… I’m afraid if these people can’t understand the basic [hygiene]… you know, go toilet and wash your hands with the soap or there’s disinfectant, use that…”

Riya from India stressed the need for cleanliness to avoid AMR infections but also noted climate change as causing environmental imbalances and changeable weather (‘sometimes it’s too much cold, sometimes it’s too much hot’) that made bodies more susceptible. Sadequa from Afghanistan, resident for 10 years, also suggested AMR was caused by ‘the weather or because of the environment? I’m not sure’.

Another theme was the need for cleanliness, with some informants extending health messages about the need for effective hand washing to speak of the need to use household disinfectants (such as Dettol) at home and the need for hospitals to be more strenuous in their cleaning, or as Aneesa suggested above, ‘spraying some [thing] around so the bugs don’t come out’. Such views encouraging the general and indiscriminate use of antimicrobials may in fact contribute to the development of resistance in the environment and intimates the difficulties in providing simple messages about appropriate actions to take to prevent AMR.

When asked about what they thought could be done about AMR, several informants subscribed to the view that the way to tackle superbugs was for science to produce new antibiotics as Kamis from Sudan suggested: ‘New ones, different to, yeah, to kill those things’. Most were shocked to learn that no new antibiotics were likely to fix the problem of resistance in the foreseeable future.

### 2.5. Travelling and Travelling Medicines

Although resident in Australia, many migrant families in our study return to their countries of origin regularly. Such travel may be for many months to visit family and friends and may include visits to local doctors and hospitals for incidental treatment for infections contracted abroad or may involve trips for deliberate consultation with local doctors within their familiar language and culture. Helena, 36 years old, travels regularly back to Sri Lanka to visit family and described one time when her husband needed extraction of wisdom teeth that they expressly sought dental care in Sri Lanka to minimize costs:

“He … he had some problem with them [wisdom teeth] coming out and then that was the time we were planning a trip to Sri Lanka [Right] and then it’s quite expensive the dental, here, the dental charges compared to there, in Sri Lanka. So we thought we’ll go and have it done there. So we went there and then got it done. Yeah, it was really cheap compared to what it is here, which is good, but then to … for the healing process, antibiotics are, yeah, recommended, prescribed. So we, we had, he had to take antibiotics.”

Q: And did the dentist explain why he needed to take them?

“I’m not sure. Probably not. Yeah, because they’re … like compared to what happens here, yeah, they [doctors in Sri Lanka] hardly explain why you, why you need something. You just get prescribed it; that’s it. That’s the process. Hardly the knowledge is shared. So he says, “Yeah, you have to take antibiotics for this many days. Take this dosage.” And that, that part is described but not why you need that. [Right] Yeah.”

Q: So is there a difference in that sense between the way in which they, antibiotics would be prescribed by a doctor here or in Sri Lanka?

“I, yeah, I think so. Normally, like not only about antibiotics but other things as well. Sri Lankan doctors generally … It has changed a bit but they have this authority of knowledge for the medicine and things like that so they hardly explain you why you need this and things like that. They just prescribe you.”

As Helena suggests, in trips back home, migrants may be prescribed or purchase antibiotics without the same level of stewardship and information. She suggested that a culture of prescribing is common among ethnic Sri-Lankan doctors in Australia and her friends prefer to seek those ‘who’ll give medicine right away’

Likewise, Hanifa, 34 years old with four children, returned back to Afghanistan for several months during the time she was resident in New Zealand. She became sick so she visited a doctor in Afghanistan and reported that she was given injections of antibiotics and tablets as: ‘The infection, the virus [in Afghanistan] is too high’. She stated that all her children in the past had received injections of antibiotics in Afghanistan and that it is possible to obtain antibiotics over the counter. Sediqua, 31 years old from Afghanistan, said that when she is asked to accompany relatives to the doctor in Australia they bring an expectation that they will receive “five or six medicines’ per ailment, like back in Afghanistan and would be angry, ‘what kind of doctor is she?’” thinking the doctor just wanted them to return and make more money by not giving them medicines.

Our study also found reference to the incidental or deliberate purchase of pharmaceuticals from overseas. In many countries, antibiotics may be purchased over the counter and stored for self-medication when needed. As Malai noted, in Thailand: ‘They just go to, to chemist and buy straight away. Don’t need prescription’. For example, one of the Dari/Pashtun interpreters at the hospital described the practices of South Asian migrants he knew who had travelled to places such as India for medical treatment:

“So they just travel back to their own country and get it [surgery] done. And of course India, they’ve got the best healthcare. There’s nothing bad about them. They’re amazing … I’ve had so many friends here that travel back to India for treatments … over there you just go to the hospital, different departments and get it done, and it’s easy. Most of them [go back to their home countries] some of them they go back home and they accumulate quite a lot of medicine and come back…”

He noted however that some migrant groups (for example migrants from Afghanistan) were reluctant to purchase medicines back home due to the fears over counterfeit medicines and the quality of medicine there. This was the case for Ying, who has lived in Australia for 22 years who said that in China medicines may not contain medication.

As the Chinese Mandarin interpreter explained: ‘Chinese people they love to take antibiotics because before, in China, you don’t they don’t even need a prescription to have antibiotics. Whenever people feel ill, they just go to the counter. You can get it over the counter. But here you need, the doctor needs to make sure’.

### 2.6. AMR Information

None of our informants cited mainstream news or television as a source of information for health matters. Most informants stated they sought health information from searches on the internet, Youtube or Facebook pages, and one German-born informant, Paula, mentioned a podcast on antimicrobial resistance as the source of her knowledge. Some preferred the internet because they could use Google translate to read information in their own languages. Many noted the need for information in community languages. Despite living in Australia for three years and before that in New Zealand for 16 years, Malika, a 36-year-old woman from Afghanistan had never heard the terms ‘AMR’, antimicrobial resistance’ or ‘superbugs’ and did not have any knowledge of resistant infections. She said she received most of her news on Facebook and hence had no knowledge of AMR. Likewise, Riya from India who has lived in Australia for over ten years stated she had never heard of ‘superbugs’ and ‘we don’t watch TV’. As a full-time housewife, with three young children she says the only news she received was via the internet. Ying from China, who has lived in Australia for 22 years, had fairly poor English and said she and her husband could not follow TV, preferring to use the internet with Google Translate so they could get a translation of the news. Another elderly Chinese background informant Mr Wang, who had lived in Australia over 20 years with reasonably good conversational English was shown a news clip explaining AMR from a public television channel. When questioned following the clip, he explained that he had not understood any of it because of the technical language and speed of delivery of the English.

Lack of English language skills may be combined with a lack of literacy. One Dari language hospital interpreter explained:

“If they’re from Afghanistan they are illiterate. They don’t have much information but they’re keen to know more… and because they can’t read or write, even if we give them flyers, if we give them information, if we print them anything from the internet, they wouldn’t be able to read it. So we ask them to ask the family to read it to them and explain.”

Another interpreter had been involved in a health information film for newly arrived Afghan refugees and suggested that this was very useful as it reached illiterate members of the community to explain public health messages. Kia from Ethiopia was adamant about the need for information in different languages:

“I think [we should] train more health professionals with a different cultural background and have, run [an information session] in a community meeting or just, you know, pass information… Every community has their own gathering and or even, actually I think the primary school and the kinder will work more when the parents drop in, the nurse can, you know, pass the information, give a flyer. Sometimes you don’t get it, even the flyers, you know. Have, have someone speak their language.”

Li from China was also keen to get information to his community:

“For migrant people, for language-limitation people, it’s very hard to get this information. And actually I, some idea come to my mind. Like I monthly I take running information session for carer support group. Yeah. Monthly, I get [a] professional from Victoria, and I think this, if someone can come up to my group to talk about these things, I think they will be very good because they, they never have … not never. They … [Chinese people] have very little opportunity to get all this like new research result and new knowledge, new, yeah, things. That’s why if someone professional they can then come to my group to talk about this. It will be good, you know.”

## 3. Discussion

Far from being passive consumers, and compliant biomedical subjects, people use medicines according to other ways of thinking and acting which may not always accord with biomedical rationalities. Cultures vary in understandings of the body, infections and causes of health and sickness, described in medical anthropology as ‘explanatory models’ [36]. Lay people may have vastly different understandings of the reasons for their sickness than that of biomedically trained professionals and often simultaneously hold what may appear to be contradictory understandings about the causality of illness. As evident in this study, informants practice syncretic care, drawing upon their lay understandings, household traditions and biomedicine in their resorts to care. Understandings of the immune system, actions of antibiotics and how antimicrobial resistance in the community develops differ, often reflecting culturally specific explanatory models mixed with information garnered from the internet, friends and medical encounters. No single generalization may be made about practices or understandings among members of any given community, each person is influenced by their cultural background and personal experiences as well as educational background and other social factors.

There remain few studies on migrant community understandings, practices and experiences of AMR despite recognition that migrants may be at higher risk of AMR infections. Our findings have implications for our understandings about how mobility intersects in various ways with medical usage, whether through the movements of people or microbes through migration or travel, or through the movements of medicine through self-importations. These have implications for the spatial assumptions underlying notions of ‘compliance’ and ‘rational drug use’ and public health interventions which tend to assume static individuals, as well as models of how resistant bacteria spread and are sustained within populations. In a globalized world in which transnational communities frequently move between countries, appreciation of the flows of people and microbes and the vulnerabilities this creates is essential.

Novel explanations of the cause of the spread of ‘superbugs’ not previously documented include climate change, dissemination by ‘new’ arrivals, and poor environmental cleanliness. This demonstrates the ways in which lay understandings adapt and constantly change and may reflect more generalized concerns within social discourse.

Humoral understandings of health remain common among our informants, particularly those from Asia and the Middle East [37]. These emerge in informants’ descriptions of sickness causality which is commonly attributed to sudden changes in body temperatures. Such humoral understandings also affect people’s understandings of the action of antibiotics and the need to take antibiotics along with other ‘cooling’ foods. However, when infections were too ‘high’, strong medicines such as antibiotics were appreciated by our informants for their efficacy, although felt to be ‘dangerous’ and heating to the body. This is consistent with findings from an in-depth qualitative study of thirty-nine households in New Zealand that found people were worried about adverse effects of antibiotics, particularly for recurrent infections. Some were concerned that antibiotics upset the body’s ‘balance’ and used strategies to try to prevent and treat infections without antibiotics [38]. In Vietnam, Craig [39] notes that antibiotics are commonly regarded as ‘fast’, ‘heavy’, ’hot’ cures which may cause a range of side effects and waste the body. Some informants avoided antibiotics for this reason, preferring to try home remedies at first signs of an infection, most considering going to the doctor only for extended or serious symptoms such as a high fever or ongoing cough that hadn’t resolved itself within a few days. Many noted differences in prescribing practices between their countries of origin and Australia.

Many informants did not differentiate between bacterial and viral infections. This may be due to a lack of English vocabulary, a lack of differentiation in the concepts in their mother tongue, or may represent a confusion of the terms and lack of health literacy, something equally true of the general Australian population. For example, in Thai the terms for viruses and bacteria are not differentiated, all are called เชื้อ *chua*, and medicine is described in terms of its action: antibiotics and anti-inflammatories are commonly called ยาแก้อักเสบ *yaa gae aksaep*—‘medicine against infection/inflammation’—as opposed to ยาแก้ปวด *yaa gae puat*—‘medicine against pain’, etc. It would not be unusual therefore for a Thai speaker not to differentiate between these terms, particularly if they were not educated in biomedical terms. This implies that health messages denoting antibiotics are only appropriate for bacterial infections may not be clear.

Our findings suggest that where and how patients procure antimicrobials must be recognized by health practitioners and policy makers. Cross-border importation of a range of pharmaceuticals and complementary medicines for self-medication is becoming a growing phenomenon in Australia—not only among migrant groups. Our study found reference to the incidental or deliberate purchase of pharmaceuticals from overseas. The extent to which these practices endanger antimicrobial stewardship is uncertain but needs to be addressed in targeted public education campaigns particularly in various community languages to improve health literacy about safe and appropriate pharmaceutical use. Self-diagnosis and self-medication is a common practice in many of our informants’ countries of origin that may lack strict antimicrobial stewardship by pharmacists or doctors. For example, a survey of 505 households in Vietnam [40] found 138 stocked drugs for anticipated illness in the future. Seventy-six households stocked a total of 96 different generic antibiotics to treat coughs or diarrhea. Antibiotics also may be consumed inadvertently in groups of medicines prescribed by medicine sellers, pharmacists or doctors which are sold in unlabeled packets. Such practices potentially expose migrants to new pathogens including resistant infections, likewise they may receive inappropriate antimicrobials or inefficient dosage. In many of our informants’ countries of origin, self-medication by purchasing over the counter or storage and inappropriate use through later self-medication may all further encourage antimicrobial resistant bacteria to develop, disseminate and be sustained in the general population.

Several informants travelled regularly and sometimes for extended periods of time visiting families and friends in their countries of origin and occasionally with the express intention of seeking medical care. Horton and Cole [41] use the term ‘medical returns’ to describe the temporary returns by immigrants to their home countries for the express purpose of seeking healthcare in their study of Mexican immigrants. Such medical returns occur either due to dissatisfaction with the health services in their countries of settlement, a desire to receive care within a more culturally and linguistically familiar setting and because immigrants may be able to afford what are perceived as ‘better’ health services ‘back home’ [41,42,43]. Cost and convenience were key considerations for making medical returns, having medicines sent from India or purchasing medicines online [44,45,46,47,48]. An Australian study [49] found that 16 out of 28 Australian-Indian migrants patients interviewed purchased a range of prescription and over-the-counter drugs and complementary and alternative medicine (CAM) from India either when travelling home; through friends and family travelling back to India; or online; without prescription and without input from a qualified health professional. Despite the fact that in Australia, pharmaceuticals are subsidized, some Indian migrant informants (for example those on student visas) reported they were not eligible for subsidized pharmaceuticals; for them purchasing back home was more cost effective.

Our study also suggests that public education campaigns through mainstream media are unlikely to reach a diverse population of people for whom English is a second language. Even among those whose English was fluent, very few people sought health information through mainstream media, preferring internet sources and Facebook groups. Education campaigns using these ‘new’ media and a variety of means, such as podcasts and films and peer to peer community training are needed in a variety of community languages [50]. However, even then, messages lacking context and clear information will do little to address the lack of understanding common to Australian society.

Very few of our informants, even those in hospital with AMR infections, were well informed about AMR or the causes for the rise in resistant bacteria. Studies suggest that health literacy about AMR in the general Australian population is low [33]. Despite a number of public health campaigns to promote appropriate use of antimicrobials and to protect from transmission of infections through hand washing there is evidence that these campaigns are relatively ineffective in reaching the general population especially within a media landscape in which multiple health conditions and messages for issue such as cancers or heart disease or climate change compete for attention and there is an increasing fragmentation of sources of information with new social media [50]. Given Australia’s multicultural diversity, it is imperative that messages regarding antibiotic stewardship be accessible for people of different backgrounds. Even migrants who have lived in Australia for many years and have relatively good English language skills may have difficulty with AMR messages and most information materials tend to be in English.

## 4. Materials and Methods

This paper is based upon interviews conducted as part of a broader community study on AMR narratives and understandings in Australia. We drew upon a subset of 31 informants with migrant backgrounds from 17 countries.

Table 2 gives a summary of the countries of origin and sex of the informants included in this paper. The sample included: 19 interviews with migrant members of the general community of differing ethnicities; seven interviews with in-patients of differing ethnic origins with AMR complications within a tertiary hospital situated in a highly diverse population in the South-east of Melbourne; and five hospital interpreting staff. This sample across a number of different ethnic groups provides insight into the experiences of people of diverse ethnicities as well as their understandings and reception of AMR information. The findings from hospital in-patients allow us to explore the experiences of people who have direct experience of an AMR infection at the time of the interview as well as the hospital staff and interpreters who deal with AMR on a daily basis. Interviews with members of the general population allow us to describe the understandings of ethnic community members and their access to information about AMR. The migrant informants interviewed in these studies represented a range of backgrounds including: recently arrived refugees with little education, tertiary-educated business migrants, people at different life stages, of different socio-economic statuses and differing employment status.

Semi-structured in-depth qualitative interviews were conducted that took approximately from one to one and a half hours. Interviews consisted of basic demographic questions followed by questions aimed at eliciting informants’ knowledge of and use of antibiotics, travel history, incidents involving treatment with antibiotics, family care, animal care and knowledge about antimicrobial resistance or superbugs. Patients with AMR infections were asked specifically about their understandings for their present hospitalization and treatment. Among the general population, interviews also included questions related to their reactions to a short video news piece on AMR to elicit their reactions and reception to the video and further questions about their information sources. All informants underwent an informed consent process and interpreters were available if required. Pseudonyms are used throughout this paper to protect their identities. Interviews were audiotaped and transcribed and thematic analysis was conducted. Ethic permissions was granted from Monash University Human Research Ethics Committee (HREC 2017-10440-12567; HREC 2018-10220-24161) and the hospital authority Monash Health as well as the hospital itself (NMA Ref: HREC/17/MonH/221; Monash Health Ref: RES-17-0000281A).

As a qualitative study, our findings are not generalizable. The study was limited due to access issues within the hospital setting (i.e., many patients with AMR infections were too ill to be interviewed). Likewise, recruiting within the community was biased as it tended to recruit only those migrants sufficiently confident in their English language skills and who wished to participate in an interview. The majority of informants were women. However, the use of face to face semi-structured interviews in both studies produced greater depth in our understanding of a range of diverse community understandings of AMR and antibiotic use than questionnaire-based community studies.

## 5. Conclusions

The findings reported here are relevant to Australia’s national antimicrobial resistance strategy which seeks to “increase awareness and understanding of antimicrobial resistance, its implications, and actions to combat it through effective communication, education and training” [51] (p. 5). Further studies into how migrant communities understand their levels of risk in relation to safe medicine use and AMR are needed. The uptake of public health messages concerning AMR requires a sensitivity to the role of cultural diversity in understandings and practices regarding antimicrobial use and the utility of anthropological and sociological approaches and theories to address this issue. Theorizing on the public uptake of scientific knowledge notes how an important factor is the epistemological status of their own ‘emic’ knowledge in relation to that of biomedicine [52]. A multidisciplinary evidence base on AMR which takes into account the context of pharmaceutical usage, social relationships involved in their use and the experiences and knowledges of diverse communities will assist health practitioners and policy makers to design and deliver targeted and sophisticated education campaigns, community-led peer interventions and regulations that can improve decision making about antimicrobial stewardship by all members of our communities.

## Figures and Tables

**Table 1 antibiotics-08-00135-t001:** Summary of lay explanations of antibiotics and antimicrobial resistance (AMR) from migrant informants with description.

Explanation	Description
Absent or Generalized	Vague or lack of knowledge of AMR or use of antibiotics; even among patients infected with AMR infections
Humoral	Antibiotics are humorally ‘hot’ and powerful. Illness caused by humoral imbalance
Antibiotics as Powerful/’Western’	Antibiotics considered strong and rapid medicines that potentially harm and weaken the body hence should be used sparingly. Home remedies preferred
Ecology and Climate	Changes in the ecology and climate facilitated the rise of more and varied bacteria and causes humoral imbalance/vulnerability. Australia as full of ‘bugs’. Bodies not suited to Australia as new environment
Mobility	Mobile populations and travel contribute to AMR (contagion theory) also related to using/purchasing medicines from overseas
‘New’ migrants	New groups bring new ‘bugs’ with them that spread to the population (contagion theory)
Hygiene	Unhygienic people who do not wash their hands (sometimes related to ‘new’ migrants) or need for clean homes and environment
Prescribing Culture and Overuse	Poor prescribing and general overuse normative in home countries. Some members of communities differ in expectations and want Australian/ethnic doctors to prescribe in a similar pattern. Mistrust doctors who do not prescribe—seen as after money
Resistant Bodies	Individual body becomes resistant; intolerant; body becomes inured to particular antibiotics
Lack of Literacy	Poor English skills means migrants cannot access information nor exposed to education messages

**Table 2 antibiotics-08-00135-t002:** Country of origin, sex and age of informants and interpreter informants.

Country of Origin	Hospital Patients	General Population	Sex	Age
Afghanistan		Aneesa	F	40
		Malika	F	36
		Hanifa	F	36
		Sadequa	F	31
Chile	Alonso		M	53
China		Ying	F	55
		Quang	F	60
		Li	M	61
		Mr Wang	M	75
India		Shanaya	F	35
		Anik	F	35
		Riya	F	37
Ethiopia		Kia	F	35
France		Rene	M	67
Germany		Paula	F	35
Pakistan		Azra	F	39
Pacific Is	Leona		F	26
Samoa	Mary		F	66
South Africa	Patricia		F	78
South Sudan		Kamis	F	38
Sri Lanka		Helena	F	36
Thailand		Malai	F	48
Netherlands	Jan		M	73
New Zealand	Peter		M	58
	Jane		F	68
Vietnam		Mai	F	32
**Total Community Informants**	7	19		26
**Hospital Interpreters** (Languages: Pashtoo/Dari, Greek, Mandarin, Urdu/Hindi, Farsi)				5
**TOTAL**				31

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
