# Peer review of "Investigating Understandings of Antibiotics and Antimicrobial Resistance in Diverse Ethnic Communities in Australia: Findings from a Qualitative Study"

_antibiotics, 2019, doi:10.3390/antibiotics8030135_

Round 1

Reviewer 1 Report

The manuscript is focusing on  the understandings of antibiotics in different ethnics communities in Australia.

The manuscript is well-prepared, but for the readers convenience it is commendable to centralise the findings in a table, highlighting the main results.

In the cases when  the informants responded that they are not using "western medicine" did they have any knowledge about the properties of the used alternatives ( garlic, ginger, coriander, etc.).

Table 1, last row, is hard to understand as long as at the first reading you may think that the reported figures are related to the first row of the table.

Author Response

We wish to thank the reviewer and have responded to all suggestions as follows: 

The manuscript is well-prepared, but for the readers convenience it is commendable to centralise the findings in a table, highlighting the main results.We have included a new Table 1 which gives a summary of major findings (line 185) In the cases when  the informants responded that they are not using "western medicine" did they have any knowledge about the properties of the used alternatives ( garlic, ginger, coriander, etc.).No they did not and we have indlcuded a sentance which indicates this (line 244).

Table 1, last row, is hard to understand as long as at the first reading you may think that the reported figures are related to the first row of the table.We have modifed the Table and belive it is now clearer

Reviewer 2 Report

The manuscript presents qualitative results resulting from a series of interviews with hospitalized patients (specifically immigrants) suffering from AMR infections often exposing beliefs and misconceptions about AMR and antibiotic uses. The paper is well written and insightful and while it does not contribute much if anything to the scientific community- science is by definition quantitative- it will undoubtedly be valuable to health practitioners and policy makers. Indeed, a critical take home message is the importance of public education campaigns designed to reach out to immigrant populations who may not speak the primary language of a given country.

I will limit my few comments to style and grammar:

Line 26: remove the word “little”.

Line 34, 46: Organism names (“Staphylococcus aureus or Escherichia coli, for example) should be italicized throughout the manuscript.

Line 79: “MRO” may be a misprint (“AMR” or “MDR”?). If not, please define.

Line 147: The introduction needs a concluding paragraph. “The purpose of this study was…”

Author Response

We thank the reviewer for their comments. Please find below our corrections:

Line 26: remove the word “little”. Thankyou, we have deleted it

Line 34, 46: Organism names (“Staphylococcus aureus or Escherichia coli, for example) should be italicized throughout the manuscript. We have changed these throughout

Line 79: “MRO” may be a misprint (“AMR” or “MDR”?). If not, please define. We have replaced with with mutiresistant organisms

Line 147: The introduction needs a concluding paragraph. “The purpose of this study was ... ” We have rearranged the introduction so that there is now a concluding paragraph as suggested. (line 163) 

Reviewer 3 Report

The manuscript "Investigating understandings of antibiotics and antimicrobial resistance in diverse ethnic communities in Australia: Findings from a qualitative study" is well structured and shows very interesting results. It contributes to a great innovative aknowledgement about the antibiotics and AMR with great originality, it takes into account many scientific aspects even including numerous social factors.

I have no comment.

Author Response

We wish to thank the reviewer for their positive assessment of our paper.